# Na^+^ Translocation Dominates over H^+^-Translocation in the Membrane Pyrophosphatase with Dual Transport Specificity

**DOI:** 10.3390/ijms252211963

**Published:** 2024-11-07

**Authors:** Alexander V. Bogachev, Viktor A. Anashkin, Yulia V. Bertsova, Elena G. Zavyalova, Alexander A. Baykov

**Affiliations:** 1Belozersky Institute of Physico-Chemical Biology, Lomonosov Moscow State University, Moscow 119899, Russia; bogachev@belozersky.msu.ru (A.V.B.); victor_anashkin@belozersky.msu.ru (V.A.A.);; 2Department of Chemistry, Lomonosov Moscow State University, Moscow 119899, Russia; zlenka2006@gmail.com

**Keywords:** energy coupling, direct coupling, proton pump, Na^+^ pump, billiard mechanism, pyranine, ETH 157, transport assay, stopped flow

## Abstract

Cation-pumping membrane pyrophosphatases (mPPases; EC 7.1.3.1) vary in their transport specificity from obligatory H^+^ transporters found in all kingdoms of life, to Na^+^/H^+^-co-transporters found in many prokaryotes. The available data suggest a unique “direct-coupling” mechanism of H^+^ transport, in which the transported proton is generated from nucleophilic water molecule. Na^+^ transport is best rationalized by assuming that the water-borne proton propels a prebound Na^+^ ion through the ion conductance channel (“billiard” mechanism). However, the “billiard” mechanism, in its simple form, is not applicable to the mPPases that simultaneously transport Na^+^ and H^+^ without evident competition between the cations (Na^+^,H^+^-PPases). In this study, we used a pyranine-based fluorescent assay to explore the relationship between the cation transport reactions catalyzed by recombinant *Bacteroides vulgatus* Na^+^,H^+^-PPase in membrane vesicles. Under appropriately chosen conditions, including the addition of an H^+^ ionophore to convert Na^+^ influx into equivalent H^+^ efflux, the pyranine signal measures either H^+^ or Na^+^ translocation. Using a stopped-flow version of this assay, we demonstrate that H^+^ and Na^+^ are transported by Na^+^,H^+^-PPase in a ratio of approximately 1:8, which is independent of Na^+^ concentration. These findings were rationalized using an “extended billiard” model, whose most likely variant predicts the kinetic limitation of Na^+^ delivery to the pump-loading site.

## 1. Introduction

Integral membrane pyrophosphatase (mPPase; EC 7.1.3.1), initially found in photosynthetic bacteria [1,2] and later in plants, protists, and many prokaryotes, catalyzes the active transport of H^+^ and/or Na^+^ out of cytoplasm [3,4,5,6,7,8,9]. The transport reaction is fueled by pyrophosphate (PP_i_), whose hydrolysis releases nearly as much energy as ATP hydrolysis (20–25 versus 35 kJ/mol) [10]. The transport function of mPPase establishes a link between chemiosmotic energy and the concentration of PP_i_, the important regulator of numerous biosynthetic reactions that produce it as a by-product [10]. In some organisms, especially plants, the reverse reaction of PP_i_ synthesis at the expense of chemiosmotic energy may also be physiologically significant [11]. mPPase-containing organisms are generally resistant to various types of stress, and this property is further enhanced in engineered mutants with increased mPPase production [12,13,14].

mPPases are divided into three families according to their transport specificities [5]. H^+^-PPases transport only protons, whereas Na^+^-PPases transport only Na^+^ at its physiological concentrations [15,16], but both Na^+^ and H^+^ at <5 mM Na^+^ concentrations [17]. In other words, the H^+^-transport activity of Na^+^-PPases is inhibited at high Na^+^ concentrations—their catalytic cycle is apparently associated with the transport of either a sodium ion or a proton, but not both. This inhibition is weak or even absent in members of the third mPPase family (Na^+^,H^+^-PPases), which transport both cations even in the presence of 100 mM Na^+^ [18,19]. All mPPases absolutely require Mg^2+^ for the transport and hydrolytic activities. K^+^ is a positive modulator of both Na^+^-transporting mPPase families and some members of the H^+^-PPase family.

The three mPPase families have very similar structures that are unique to the protein world [20,21,22]. All mPPases are homodimers of approximately 70-kDa α-helical subunits. Each subunit forms a funnel-like structure with a hydrophilic cavity (hydrolytic center) in the cytoplasmic part. The cavity contains protein ligands for five Mg^2+^ ions and pyrophosphate and nucleophilic water molecules. A gated exit channel connects the hydrolytic center to the other side of the membrane (vacuolar lumen or periplasmic space). The major difference between Na^+^-PPase and H^+^-PPase is the channel position of the conserved Glu residue that forms a Na^+^-binding site in the Na^+^-PPase [23].

The structural similarity of the three mPPase subfamilies suggests a common coupling mechanism. H^+^-PPase is structurally poised to “direct coupling”, wherein the transported proton is generated by the nucleophilic water molecule during its attack on PP_i_ in the catalytic site (the “chemical” proton) and is translocated to the adjacent exit channel via short proton wires linking auxiliary proton-binding sites [20]. Direct coupling appears to be a cornerstone of the H^+^ translocation mechanism in both H^+^-PPase [3,20] and the two Na^+^-transporting mPPase subfamilies when they function as H^+^ transporters [3]. This H^+^ transport mechanism provides a simple venue to the Na^+^ transport via creating an intermediate Na^+^-binding site on the proton path through the channel (Figure 1A). Such a site formed by the aforementioned Glu residue was indeed found in the crystal structure of Na^+^-PPase [21]. In the “billiard” mechanism of the Na^+^ transport [3], the water-borne proton electrostatically pushes the bound Na^+^ ion through the exit channel. In the framework of this mechanism, the dual specificity of Na^+^-PPase at low Na^+^ concentrations is explained by the ability of the Na^+^-binding site to competitively accommodate the Na^+^ and proton (Figure 1B).

However, the “billiard” mechanism, in its simple form, cannot explain the transport specificity of the so-called Na^+^,H^+^-PPases that transport H^+^ along with Na^+^ even in the presence of 100 mM Na^+^, suggesting that the two transport reactions are independent [18,19]. Two hypothetical explanations have been proposed—one assumes that both transport reactions occur in the same subunit [3], while the other posits that they occur in different subunits within a dimer [22]. Much uncertainty about the pumping mechanism of Na^+^,H^+^-PPase could be resolved by measuring the H^+^/Na^+^ pumping ratio/stoichiometry. However, this important parameter has remained unknown because previous transport assays of the two cations were based on different principles—^22^Na^+^ uptake and ΔpH indicator fluorescence change, not allowing quantitative comparisons.

We describe here an indirect pyranine-based assay that allows the separate monitoring and quantification of both transport reactions on a millisecond scale. By using the assay with homodimeric *Bacteroides vulgatus* Na^+^,H^+^-translocating mPPase (*Bv*-mPPase), we show that the Na^+^/H^+^ transport ratio is much less than unity under a variety of conditions. This finding rules out the simple competition model of Na^+^ and H^+^ transport via the same protein machinery and transport mechanism, suggesting a unique coupling mechanism for this relict ion pump.

## 2. Results

### 2.1. The Unified Procedure to Measure Na^+^ and H^+^ Transport into Membrane Vesicles

mPPase from *B. vulgatus* was the first characterized Na^+^,H^+^-PPase capable of simultaneous translocation of Na^+^ and H^+^ over a wide range of their concentrations in the medium [18]. Knowing the ratio of the Na^+^ and H^+^ transport rates is important for the diagnosis of Na^+^,H^+^-PPase mechanism. A ratio close to unity would indicate the asymmetrical functioning of transporter subunits, such that one subunit transports Na^+^ and the other transports H^+^, as proposed in a recent publication [22]. On the other hand, the prevalence of one activity would favor the transport promiscuity of both subunits.

Therefore, we developed a unified procedure for estimating the rates of concurrent Na^+^ and H^+^ transport into inverted membrane vesicles harboring *Bv-*mPPase in the membrane and pyranine in the lumen. This pH indicator fluoresces only in a deprotonated form, allowing one to monitor H^+^ translocation through the vesicle membrane in both directions [24,25,26]. The advantage of pyranine as a pH indicator over the more common acridines is that it reports on changes in pH in the aqueous phase rather than ΔpH across the membrane and its response is therefore more linear. The pyranine response is bidirectional, less prone to artefact formation, and much more rapid because it does not require an indicator transport through the membrane.

The principle of the assay is illustrated in Figure 2. *Bv-*mPPase establishes fluxes of Na^+^ and H^+^ into vesicles produced from recombinant *E. coli* cells (Figure 2). If the Na^+^ ionophore ETH 157 is present, it will return the Na^+^ ions to the outer medium (blue arrow in Figure 2A). Furthermore, the Na^+^ ionophore allows the translocation of additional Na^+^ ions (red arrow in Figure 2A) to compensate for the charge coming with the translocated H^+^ ions. This helps to avoid transport rate limitation by the transmembrane electric potential difference Δψ. The total rate of passive Na^+^ transport by ETH 157 is equal to the sum of the rates of Na^+^ and H^+^ pumping by *Bv-*mPPase, resulting in *acidification* (H^+^ accumulation) inside the vesicles. In the presence of ETH 157, the pyranine fluorescence reports thus on H^+^ pumping into the vesicles with no interference from the Na^+^ transport, which does not affect the vesicle acidity directly. According to the pyranine titration curve, a 10% change in the pyranine fluorescence will result from a pH change of approximately 0.2 unit. In a suspension of pyranine-loaded vesicles, the same pH change in the vesicle lumen is expected to cause a smaller relative change in the measured fluorescence because of the background signal from the contaminating pyranine in the medium.

Similarly, pyranine fluorescence in the presence of the protonophore carbonyl cyanide *m*-chlorophenylhydrazone (CCCP) can monitor Na^+^ translocation by *Bv-*mPPase with no interference from the H^+^ transport [27,28]. In this setup, the H^+^ ions pumped in return to the outer medium through CCCP (red arrows in Figure 2B). The protonophore also discharges the Δψ formed due to Na^+^ pumping in by *Bv-*mPPase. In a medium lacking membrane-permeable ions other than H^+^, the rate of proton release from membrane vesicles will equal the rate of Na^+^ pumping in by *Bv-*mPPase (blue arrows in Figure 2B) [29]. Accordingly, *Bv-*mPPase functioning in the presence of CCCP will *alkalinize* the vesicle lumen.

Importantly, the transport measurements reported below were performed at pH 7.2, which is equal to pyranine p*K*_a_ [25], and the vesicles contained large amounts (100 mM) of a Mops buffer with the same p*K*_a_. With this experimental setup, equal fluxes of H^+^ and Na^+^ ions into the vesicles should result in symmetrical fluorescence changes caused by alkalinization in the presence of CCCP or acidification in the presence of ETH 157 (Figure 2C). This evident prediction was confirmed in trial titrations of a pyranine-loaded vesicle suspension with alkali or acid in the presence of 2 µg/mL nonspecific ionophore gramicidin D. The sensitivity of the assay is determined by the slope of the tangent to the pyranine titration curve, *dF*/*dpH*, and is maximal at pH = pyranine p*K*_a_ (Figure 2C). Although this assay does not yield absolute transport rates, it allows the comparison of the Na^+^- and H^+^-translocating activities of Na^+^,H^+^-PPase under identical conditions in the same vesicle preparation.

The validity of the proposed approach was tested with the H^+^-translocating mPPase of *Desulfitobacterium hafniense* (*Dh-*mPPase) and Na^+^-translocating mPPase of *Desulfuromonas acetoxidans* (*Da-*mPPase) produced in *E. coli* cells. In the presence of 10 mM Na^+^, these mPPases are strictly cation-specific—*Dh-*mPPase transports only H^+^, whereas *Da-*mPPase transports only Na^+^ [15,30]. Notably, the membrane vesicles produced by the procedure used were predominantly inverted (>90%), as estimated from the activating effect of alamethicine on their NADH oxidase activity [31] (Appendix A).

As shown in Figure 3A, the PP_i_ addition to *Dh-*mPPase-containing vesicles progressively decreased pyranine fluorescence, i.e., acidified the vesicle lumen. Keeping in mind that the vesicle membrane is impermeable to PP_i_ and that lumen acidification did not occur with the vesicles prepared from non-transformed *E. coli* cells and was completely abolished by the Na^+^(K^+^)/H^+^-exchanger monensin, one can infer that the pyranine signal reports on the H^+^-transporting activity of *Dh-*mPPase. This conclusion was consistent with observations that the signal did not depend on the medium Na^+^ concentration and was accelerated by the K^+^ ionophore valinomycin and Na^+^ ionophore ETH 157 (Figure 3A), which collapsed *Dh-*mPPase-generated Δψ. CCCP suppressed the pyranine signal (Figure 3A) as expected, by facilitating the H^+^ exit from the vesicles.

In contrast, the Na^+^-translocating *Da-*mPPase caused monensin-sensitive alkalinization of the vesicle lumen in response to PP_i_ addition (Figure 3B) because the Na^+^ accumulation stimulated passive H^+^ leakage from the vesicles. CCCP increased the rate of fluorescence change by increasing the membrane permeability to protons. The rate increased monotonically with the Na^+^ concentration in the range of 0.05–50 mM. The Na^+^ ionophore ETH 157 and K^+^ ionophore valinomycin suppressed the pyranine signal appreciably (Figure 3B) by allowing Δψ-driven Na^+^ or K^+^ ions to exit from the vesicles instead of proton. The evident corollary is that the proposed transport assay adequately measures the Na^+^- and H^+^-translocating activities of the monospecific H^+^- and Na^+^-PPases.

### 2.2. Na^+^ and H^+^ Fluxes Mediated by Dual-Specificity Bv-mPPase

PP_i_ addition to the membrane vesicle harboring the promiscuous Na^+^,H^+^-translocating *Bv-*mPPase caused monensin-sensitive alkalinization of the vesicle lumen in the medium containing 10 mM Na^+^ and no ionophores (Figure 4). Lumen alkalinization was markedly accelerated by CCCP, indicating low membrane permeability to protons in the absence of a protonophore. The pyranine response was only partially decelerated by valinomycin, which collapsed the Δψ. These findings were the first to indicate a marked prevalence of the Na^+^-translocating activity of *Bv-*mPPase over its H^+^-translocating activity—zero signal would be observed if the activities were equal. When added together, the ionophores ETH 157 and CCCP markedly suppressed lumen alkalinization (Figure 4). Notably, H^+^-translocation by *Bv-*mPPase (fluorescence decrease) was detected only in the presence of ETH 157, which neglected the contribution of the Na^+^-translocating activity to the pyranine signal (Figure 2A). But even in this case, the acidification of the vesicle lumen proceeded markedly slower than its alkalinization in the presence of CCCP (Figure 4). Clearly, the simultaneous fluxes of Na^+^ and H^+^ mediated by *Bv-*mPPase differ markedly in intensity, with the Na^+^ flux prevailing.

To probe the direct effects of the ionophores on the *Bv-*mPPase, the time-courses of the P_i_ formation from 150 µM PP_i_ by *Bv-*mPPase-containing vesicles were obtained using a continuous P_i_ analyzer [32]. The effects of ETH 157 and valinomycin under the conditions shown in Figure 4 were insignificant (<5%) and CCCP stimulated the hydrolysis reaction by only 20% (Appendix A). These findings rule out the direct interaction of the ionophores with *Bv-*mPPase as the only explanation of their effects on the cation transport.

### 2.3. Fast Kinetics of Cation Transport by Bv-mPPase: Effects of Na^+^ Concentration

Quantitative estimates of the initial rates of the cation transport by *Bv*-mPPase could not be obtained with precision from the data presented in Figure 4, which were measured using manual mixing. Therefore, Na^+^ and H^+^ transport measurements in the presence of CCCP and ETH 157, similar to those in Figure 4, were performed using a stopped-flow instrument with a mixing time of 1.6-ms. Figure 5 shows typical recordings obtained at three Na^+^ concentrations. Signals corresponding to either the Na^+^ or H^+^-translocation were observed over the entire range of the Na^+^ concentrations, consistent with earlier findings [18]. The smaller signal amplitudes than in Figure 4 are explained by the higher contribution of the light scattered by the membrane vesicles to the signal because of the inferior performance of the filter used to cut off the exciting light in the stopped-flow instrument compared with the monochromator in the conventional spectrofluorometer.

The time courses of the pyranine fluorescence in Figure 5 could be reasonably well described by a one-exponent Equation (1) or (2) (see Section 4), yielding, in each case, the absolute value of the initial velocity of the fluorescence change (*v*_0_, a measure of the transport rate) and the conditional rate constant *k* (Appendix A). The latter parameter does not have a physical meaning and is used only as a means to fit the data and determine the *v*_0_ value. The Na^+^-translocating activity measured with the CCCP (*v*_0_^Na^) and H^+^-translocating activity measured with ETH 157 (*v*_0_^H^) increased approximately in parallel with the increase in [Na^+^] (Figure 6A). However, the major finding was that the ratio *v*_0_^H^/*v*_0_^Na^ was markedly less than unity and nearly constant in the 1–50 mM Na^+^ concentration range (Figure 6B). Based on the *v*_0_^H^/*v*_0_^Na^ values obtained (Appendix A and Figure 6B), H^+^ influx contributes only ~1/8th to the total monovalent cation influx catalyzed by *Bv*-mPPase in the Na^+^ concentration range tested.

Notably, the Na^+^ dependence of the hydrolytic activity (Figure 6C) differed from the dependences of both transport activities (Figure 6A). The former dependence was a simple hyperbola, consistent with a single essential Na^+^-binding site with *K*_d_ of 0.40 ± 0.05 mM (earlier reported as 0.57 mM [18] under similar conditions), nearly saturated at 10 mM Na^+^. In contrast, the transport activities required higher Na^+^ concentrations for saturation (Figure 6A), suggesting a requirement for an additional, lower-affinity Na^+^-binding site.

The lower absolute rate of the fluorescence change due to the H^+^ transport (ETH 157 present) compared with the Na^+^ transport (CCCP present) in Figure 4, Figure 5 and Figure 6 might have resulted from the kinetic limitation of the ETH 157-mediated Na^+^ efflux. Notably, the standard ETH 157 concentration used (20 µM) is near the solubility limit and could not be further increased. However, several lines of evidence ruled out the possibility of H^+^ transport rate limitation by the Na^+^ efflux through ETH 157. First, adding 1 µM valinomycin to the assay medium with ETH 157 to allow K^+^ efflux and decrease the burden of ETH 157 did not accelerate the initial rate of fluorescence change (H^+^ transport) at the 10 mM Na^+^ concentration. Second, a twofold decrease in the ETH 157 concentration resulted in only a 20% decrease in *v*_0_^H^ at a 10 mM Na^+^ concentration, suggesting that the 20 µM ETH concentration was nearly saturating. Finally, a higher rate of ETH 157-stimulated transport observed for the different mPPase in Figure 3A also makes it unlikely that the transportation capacity of ETH 157 was exhausted in Figure 4 and Figure 5.

### 2.4. Fast Kinetics of Cation Transport by Bv-mPPase: Effects of Substrate Concentration

The preceding hydrolysis and transport measurements were conducted at a constant 150 µM PP_i_ concentration. Figure 7 details the effects of substrate concentration on hydrolysis and transport rates. Because mPPase activities depend on the concentration of the Mg^2+^ cofactor, its free concentration in the medium was maintained at a constant level of 4.8 mM in these experiments by taking into account its complexation with PP_i_ (see Section 4).

The hydrolysis rate profile shown in Figure 7A was similar to that measured earlier under similar conditions [30]. The profile is of an asymmetrical bell type—its rising and descending parts correspond to substrate binding to the first and second catalytic sites in the mPPase dimer (Figure 1) [22,30,33,34].

Fitting Equation (3) (see Section 4) to the rate dependence in Figure 7A yielded the values for the corresponding macroscopic Michaelis constants (*K*_m1_ and *K*_m2_) and the maximal activities of the mono- and di-substrate complexes (*A*_1_ and *A*_2_) listed in Table 1. Notably, the *A*_1_ and *A*_2_ values refer to one and two functioning active sites in dimer, respectively; hence, the actual average performance of each of the two active sites in SES was approximately one third of that of the productive subunit in ES. The macroscopic *K*_m1_ value previously measured for *Bv-*mPPase under similar conditions was 12 ± 1 µM in terms of the Mg_2_PP_i_ complex [30] (corresponding to 18 µM in terms of the total PP_i_, as used in this work). Figure 7B shows the distribution of the enzyme between the two enzyme–substrate complexes at different substrate concentrations, as calculated based on the *K*_m_ values obtained in this work.

Stopped-flow transport measurements were additionally performed at a 10 µM PP_i_ concentration (Appendix A), and the kinetic parameters were derived from the time courses as described above (Appendix A). Like the effects of Na^+^ on the Na^+^ and H^+^ transport functions of *Bv*-mPPase, the effects of PP_i_ were symmetrical (Figure 7C,D)—the Na^+^ and H^+^ transport rates changed in parallel when the PP_i_ concentration was varied. By comparing the effects of the PP_i_ concentration on the rates of hydrolysis and the Na^+^/H^+^ transport and the amounts of two enzyme-substrate complexes, one can conclude that all the activities are primarily associated with the ES complex, with a lesser or no contribution of the SES complex. These data also rule out hypothetical model(s) in which a particular transport function is associated with only one of these complexes.

## 3. Discussion

Transport promiscuity is not uncommon among cation transporters, including those active with the sodium ion. Na^+^,K^+^-ATPase [35], Na^+^-coupled rotary ATP synthases [36,37,38], and Na^+^-rhodopsin [39] can transport H^+^ instead of Na^+^ at low pH and/or Na^+^ concentrations. *Methanosarcina acetivorans* A_1_A_0_-ATP synthase is unique in that it is capable of generating/consuming both Na^+^ and H^+^ gradients under physiological conditions, suggesting no competition between the cations [40]. Notably, all ATPases use an indirect-coupling mechanism, in which the transported protons reach the pump-loading site (located in c-subunit of the rotary ATPases) from the medium. A suitable modification of this site can change its cation-binding specificity and, hence, pump specificity [41].

mPPase differs from rotary ATPases in that its H^+^ transport function is based on a “direct coupling” principle—the transported proton is generated from one of the reactants, the nucleophilic water molecule. This mechanism was proposed by Lin et al. based on the structure of the H^+^-transporting mPPase, which demonstrated a close contact between the active site with the bound PP_i_ analog and the transport channel [20]. In the mPPase structure, the water nucleophile is located at the channel entry and is coordinated by two Asp residues [20], as in aspartic proteases [42]. This coordination positions the water molecule for attack and increases its nucleophilicity, but is insufficient to convert water into a hydroxide, unlike in soluble PPases, in which the water molecule is coordinated by three divalent metal ions [43] or two such ions and Asp [44]. The Asp-only coordination results in a much lower hydrolysis efficiency of mPPases but permits proton generation from the nucleophilic water molecule in the proper place for its subsequent transport.

Strong functional evidence for the “direct coupling” mechanism is found in the data of Li et al. [45] and Shah et al. [46], who directly measured the charge flux through H^+^-translocating mPPase embedded in lipid bilayer. They observed a small signal due to the binding of the nonhydrolyzable PP_i_ analog imidodiphosphate, which they interpreted as a binding-induced proton transfer. They also observed a 10-fold greater signal generated by PP_i_, which they ignored, perhaps because it was not recognized at that time that the latter signal resulted from a single turnover of the PP_i_ hydrolysis [3,47]. Simple logic tells us that if a complete turnover yields a 10-fold greater signal than the substrate binding, the latter event is not enough for a proton to cross the membrane. Clearly, substrate binding may cause a proton or other charge to cross 1/10 of the membrane thickness, but the complete translocation event (proton disappearance on one side of the membrane and appearance of the same or a different proton on its other side) requires a complete hydrolysis cycle. This seminal result unambiguously rules out substrate binding as the step at which proton translocation occurs, but nevertheless, publications from Goldman’s group keep propagating this idea [22,23,48]. Notably, mPPase is the only non-oxidoreductase directly coupled proton pump—all other such pumps transport electrons [49]. Rhodopsin uses a similar mechanism to pump the proton but differs in that the transported proton appears due to a light-induced shift in retinal Schiff base p*K*_a_, not via a chemical reaction as it is in mPPase [50].

Na^+^ ion is not a reaction product and cannot be transported by mPPase in the same way as H^+^. However, given the very similar architecture of Na^+^- and H^+^-pumping mPPases, it is highly unlikely that the “chemical” proton is not a factor in the Na^+^ transport [34]. The “billiard” hypothesis described in the Introduction provided a simple way to integrate the two transport functions [3,5,47]. The hypothesis is based on structural data showing a Na^+^-binding site in the ion conductance channel of the Na^+^-transporting mPPase from *T. maritima* [23,45] and phylogenetic analyses showing its conservation in related mPPases. The principal Na^+^ ligand at this pump-loading site is a Glu residue. Although its affinity for protons is much greater, it is Na^+^-bound under physiological conditions simply because of a much greater Na^+^ concentration in comparison with the H^+^ concentration. After dislodging the bound Na^+^ ion into the exit channel, the “chemical” proton can occupy the Glu site and be transported in the next catalytic cycle or displaced into the cytoplasm by an incoming Na^+^ ion. The choice between these possibilities depends on the Na^+^ concentration, and when it is high enough, the H^+^ transport should come to a stop, provided that the Na^+^ binding to the Glu can completely prevent the “chemical” proton binding. This is possible if the Na^+^ binding is a second-order process, whose rate tends to infinity with the increasing Na^+^ concentration. Such behavior is characteristic of typical Na^+^-transporting mPPases [17].

The deviation of Na^+^,H^+^-transporting mPPases from this rule may have several explanations. Li et al. [45] and Strauss et al. [22] proposed that Na^+^ and H^+^ are alternately transported by Na^+^,H^+^-PPase subunits, which flip between different ion gate configurations—it corresponds to H^+^-PPase in one subunit, and to Na^+^-PPase in the other. They also hypothesized that the transported cation binding to subunit B is necessary for product release from subunit A. However, this mechanism suggests equal rates of Na^+^ and H^+^ transport and is therefore inconsistent with our data.

A more likely alternative is that both transport events occur in the same subunit and the well-documented non-equivalence of subunits [22,23,30,33,51] has a different explanation. For instance, subunit A may perform hydrolysis and transport while subunit B transiently stores conformational energy, or subunits work alternately—PP_i_ binding to subunit B is necessary for product release from subunit A [3].

To our knowledge, the transport stoichiometry of the Na^+^,H^+^-PPase may have three explanations. One possibility (Model A) is that Na^+^ binding to the pump-loading site involves two steps: transient binding to an auxiliary site (step 1, a second-order reaction with the rate increasing with the Na^+^ concentration) followed by the Na^+^ transfer to the destination (step 2, a first-order reaction with the rate independent of the Na^+^ concentration). The overall binding rate then increases with the increasing Na^+^ concentration to a constant level equal to the rate of step 2. In this working model, the limiting distribution of the Glu residue between H^+^-bound and Na^+^-bound forms at high Na^+^ concentrations is determined by the relative rates of Na^+^ delivery from the auxiliary site (step 2) and H^+^ delivery from the transition state complex. Accordingly, a constant ratio of the H^+^ and Na^+^ transport rates is expected at high Na^+^ concentrations. In principle, this ratio may be smaller or greater than one, and our data (Figure 6) show that it is approximately 0.14 for the *Bv*-mPPase.

That this ratio tends to zero in Na^+^-transporting mPPases suggests that they lack a transitory site or that it delivers Na^+^ to the pump-loading site at a much higher rate compared with H^+^. The available data do not allow a rigorous choice between these alternatives. Although the published crystal structures of the Na^+^-transporting mPPase from *T. maritima* revealed a single pump-loading Na^+^-binding site per subunit [23,45], functional data indicated the presence of an additional, Na^+^-specific high-affinity site, whose occupancy by Na^+^ is required for PP_i_ hydrolysis and H^+^/Na^+^ transport [17,24,52]. Given that the crystal structure does not necessarily reveal all binding sites, the “two Na^+^ sites” model is preferable, but further work is clearly needed, including a determination of the structure of Na^+^,H^+^-PPase in a Na^+^-bound state.

Alternatively, the “chemical” proton may partially reverse Na^+^ binding to the pump-loading site, which becomes a bifurcation point in the Na^+^ pathway in this case (Model B). In other words, there may be a non-zero probability that the released “chemical” proton directs a fraction of bound Na^+^ ions back to the cytoplasm or an alternative location and partially occupies the pump-loading site. In this case, the H^+^/Na^+^ transport ratio may also decrease to a constant level with the increasing Na^+^ concentration.

Models A and B imply the transport stoichiometry of approximately 0.9 Na^+^ and 0.1 H^+^ ion, summing up to 1 ion per each hydrolyzed PP_i_ molecule. Because our data estimate only the ratio of transport rates, they permit the stoichiometry of 1 Na^+^ and 0.11 H^+^ ions per PP_i_ molecule hydrolyzed, giving more than one cation in total. This could happen if the “chemical” proton could enter the exit channel together with Na^+^ in a fraction of catalytic cycles, provided that the time during which the channel is open permits this (Model C). In Model C, the transportations of Na^+^ and H^+^ are thus intrinsically coupled in the catalytic cycle, whereas they are competing processes in Models A and B. The independence of the H^+^/Na^+^ transport ratio on the Na^+^ concentration in Model C may have the same explanation as in Model A or B.

To summarize, the results reported above provide functional and kinetic insights into the ion transport mechanisms underlying the dual transport specificity of pyrophosphate-driven Na^+^ pumps. Notably, Na^+^,H^+^-PPases are abundant in gut bacteria and considered to represent a means of adapting to environmental challenges (anaerobiosis and intense niche competition) [18]. Further work is clearly required to reliably choose between the transport and hydrolysis models mentioned above and to delineate the general transport mechanism in sufficient detail to explain the transport promiscuity of mPPases. This work will likely involve site-directed mutagenesis along with the aforementioned assays and time-resolved structural studies using direct methods and molecular dynamics simulations.

## 4. Materials and Methods

### 4.1. Materials

Pyranine was obtained from Eastman Kodak (Rochester, NY, USA); Tris (Trizma base), valinomycin, and carbonyl cyanide *m*-chlorophenylhydrazone (CCCP) were obtained from Sigma-Aldrich Co (St. Lous, MO, USA); Mops was an Amresco (Solon, OH, USA) product; the Na^+^ ionophore ETH 157 and K_2_SO_4_ were from Fluka Chemie (Buchs, Switzerland); magnesium sulfate hexahydrate was from Merck (Rahway, NJ, USA); disodium sulfate was from Reachim (Moscow, Russia); monensin was obtained from Serva-Feinbiochemica (Heidelberg, Germany). 

### 4.2. Production of mPPases in Escherichia coli and Isolation of Membrane Vesicles

Plasmids with genes for mPPases from *B. vulgatus* (*Bv*-mPPase) [17], *Desulfitobacterium hafniense* (*Dh*-mPPase) [30], and *Desulfuromonas acetoxidans* (*Da*-mPPase) [15] were expressed in *E. coli* C41(DE3) cells as described elsewhere [30]. *E. coli* cells were harvested via centrifugation (10,000× *g*, 10 min) and washed with medium containing 85 mM NaCl, 5 mM MgSO_4_, and 10 mM Tris-HCl, pH 7.5. The sediment was suspended in medium A (100 mM Mops-KOH, pH 7.2, 25 mM K_2_SO_4_, 5 mM MgSO_4_) containing 1 mM pyranine and traces of DNase. The mixture was passed once through a French press at 16,000 psi, the unbroken cells and cell debris were removed by centrifugation at 27,500× *g* (5 min), and membrane vesicles were sedimented at 150,000× *g* (50 min). The vesicles were additionally washed with medium A via two cycles of resuspension/centrifugation (150,000× *g*, 40 min) and immediately used for transport and hydrolytic activity measurements. Vesicles were quantitated in terms of their protein concentration using a bicinchoninic acid method [53] with bovine serum albumin as a standard. mPPase activities of the membrane vesicles harboring *Bv-*mPPase, *Dh-*mPPase, or *Da-*mPPase were 0.34, 1.2, and 0.60 μmol·min^−1^·mg protein^−1^, respectively. Sodium fluoride (1 mM) was added to the activity assay to inhibit traces of contaminating *E. coli* cytosolic PPase.

### 4.3. Assay of the Hydrolytic Activity

The rates of PP_i_ hydrolysis were measured at 25 °C using a continuous P_i_ assay [32] with a sensitivity of 6 μM P_i_ per recorder scale. The reaction mixture of 20 mL volume typically contained 100 mM Mops-KOH buffer, pH 7.2, 5 mM MgSO_4_, 25 mM K_2_SO_4_, 5 mM Na_2_SO_4_, and membrane vesicles (2–16 μg protein/mL). The reaction was started by the addition of 150 μM PP_i_ (tetrasodium salt), and P_i_ liberation was monitored for 3–4 min. Variations in the assay composition are specified in figure legends. All activity measurements were performed in duplicate, and appropriate corrections were made for PP_i_ interference with P_i_ assay [33]. Hydrolysis rates (mPPase activities) are presented below in terms of the molar amount of PP_i_ hydrolyzed per 1 min per 1 mg of vesicle protein.

### 4.4. Assay of the Transport Activities

The acidification and alkalinization of the vesicle interior were monitored by measuring the fluorescence of the entrapped pyranine at 510 nm with excitation at 458 nm in medium A containing different concentrations of Na_2_SO_4_. The contaminating Na^+^ concentration in medium A was approximately 50 μM, as measured using a flame photometer (UNICO-SYS, Saint Petersburg, Russia). Fluorescence measurements were performed using a FluoroMax-3 spectrofluorometer (Horiba Jobin Yvon, Stow, MA, USA). The vesicles were preincubated for 5 min in the assay medium before adding 150 μM PP_i_ (tetrapotassium salt). Other additions to the assay medium are described in the figure legends. PP_i_ and monensin were added from their respective 100 mM and 2.5 mM stock solutions.

Stopped-flow measurements of the fluorescence time courses upon PP_i_ addition to the membrane vesicles containing entrapped pyranine were performed on a rapid kinetics apparatus (BioLogic Science Instruments, Seyssinet-Pariset, France) consisting of an SFM-3000/S stopped-flow mixer and MOS-200 optical system with a 3-µL cuvette. The excitation wavelength was set at 458 nm (monochromator slit width of 4 nm) and emission was detected at >495 nm using a SCHOTT GG495 filter (Mainz, Germany). Equal volumes (13 µL each) of the vesicle suspension and PP_i_ solution in the same buffer were mixed at 25 °C at a flow rate of 1.2 mL/s (dead-time of 1.6 ms), and fluorescence was monitored for 20 s at a 10-ms resolution. Data from approximately 20 shots were averaged in each case.

### 4.5. Data Treatment

All nonlinear least-squares data fittings were performed using Scientist software version 2.01 (MicroMath, Salt Lake City, UT, USA). Stopped-flow data were analyzed in terms of simple first-order kinetics using Equation (1) (fluorescence rise) or 2 (fluorescence decay), where *F* is the fluorescence, *v*_0_ is the initial rate of fluorescence change, *k* is the rate constant, *t* is time, and *a* is *F* offset; the ratio *v*_0_/*k* is equal to the amplitude of the signal change.
(1)$$F=a+(v_{0}/k)\cdot \left[1-\exp\left(-kt\right)\right]$$
(2)$$F=a+(v_{0}/k)\cdot \exp\;\left(-kt\right)$$

Cooperative kinetics of PP_i_ hydrolysis was analyzed in terms of Figure 1 [30], where *K*_m1_ and *K*_m2_ are the Michaelis constants for consecutive PP_i_ (S) binding to two active sites in the enzyme dimer (E); *A*_1_ and *A*_2_ are the specific rates for the ES, and SES species, respectively. The corresponding rate equation (Equation (3)) implies steady-state kinetics for substrate binding and conversion. The scheme and equation(s) describe hydrolysis kinetics in terms of macroscopic Michaelis constants, i.e., do not distinguish between the two ES species containing substrate in different subunits of the dimer.
*v* = (*A*_1_ + *A*_2_[S]/*K*_m2_)/(1 + *K*_m1_/[S] + [S]/*K*_m2_)(3)

All kinetic measurement reported above were performed at a free Mg^2+^ concentration of 4.8 mM in the reaction media. To accomplish this, we calculated the amounts of the Mg^2+^ ions bound in MgPP_i_ and Mg_2_PP_i_ complexes formed at each PP_i_ concentration using the published data on the equilibria in the Mg^2+^–PP_i_ system [32] and increased the amount of the added magnesium salt appropriately. In the presence of 4.8 mM Mg^2+^ ions, 63% of the total PP_i_ exist as the Mg_2_PP_i_ complex, the presumed true substrate of mPPase, at pH 7.2 [32].

## Data Availability

The data presented in this study are available on request from the authors.

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
