# Peer review of "Na+ Translocation Dominates over H+-Translocation in the Membrane Pyrophosphatase with Dual Transport Specificity"

_ijms, 2024, doi:10.3390/ijms252211963_

Round 1
Reviewer 1 Report
Comments and Suggestions for Authors
1. Figure 1 should be (A) in replace of A in the caption and same as for the each figure caption..
2. In Figure 2, there is written caption “A) Ionic fluxes through the vesicle membrane (yellow circle)” please write the specific color, the circle is not yellow.
3. In this manuscript, There should be one clear mechanism for understanding the Na and H ion transportation.
4. For the explanation of Figure 4, Lumen alkalinization is mentioned please refer to the reference.
5. In Figure 5, Please use the subscript M K2SO4, 5 mM MgSO4, as K2. Same for the other chemical formulae.
6. Please specify the hydrolytic measurement conditions for Figure 6C.
Author Response
We thank the reviewer for his/her time. We describe below the changes made in response to the remarks or explain why we politely decline them. Modified or added text is highlighted in the revised manuscript.
- Figure 1 should be (A) in replace of A in the caption and same as for the each figure caption.
RESPONSE: Please note that both styles of figure labelling are used in the latest published IJMS issue, with the one we use being more common.
- In Figure 2, there is written caption “A) Ionic fluxes through the vesicle membrane(yellow circle)” please write the specific color, the circle is not yellow.
RESPONSE: To be more exact, we have replaced “yellow” with “dark yellow”. This is exactly the color we see on the monitor screen.
- In this manuscript, There should be one clear mechanism for understanding theNa and H ion transportation.
RESPONSE: Such a mechanism is the ultimate goal of our research, and we hope to establish it in our future studies. At this stage of the research, we cannot conclusively formulate this mechanism and prefer to formulate several hypothetical mechanisms consistent with the available data, rather than to focus indiscriminately on one of them.
- For the explanation of Figure 4, Lumen alkalinization is mentioned please refer to the reference.
RESPONSE: To further clarify the description of lumen alkalinization in the first sentence of this paragraph, we have added “and no ionophores” in line 210.
- In Figure 5, Please use the subscript M K2SO4, 5 mM MgSO4, as K2. Same forthe other chemical formulae.
RESPONSE: Corrected.
- Please specify the hydrolytic measurement conditions for Figure 6C.
RESPONSE: Please note that the conditions are fully detailed in section 4.3 of Materials and Methods.
Reviewer 2 Report
Comments and Suggestions for Authors
The paper addresses Na+ translocation dominates over H+-translocation in the membrane pyrophosphatase with dual transport specificity.
The research gap is clearly stated as the authors tackle that Na+ /H+ transport ratio is much less than unity under a variety of conditions. However, the direct application for this research is not specified.
Results and discussion:
The authors refer to the term "chemical" proton which is incorrect.
Lines 173-175. The membrane vesicles produced by the procedure used were predominantly inverted (> 90 %), as estimated fromhe activating effect of alamethicine on their NADH oxidase activity [31] (data not shown). Why is this not shown?
Kinetics parameters for Table 1 must be previously explained or mentioned in other sections.
Quality of the figures and tables is very good and easy to follow.
Authors do not include conclusions.
Author Response
We thank the reviewer for his/her time. We describe below the changes made in response to the remarks or explain why we politely decline them. Modified or added text is highlighted in the revised manuscript.
- The authors refer to the term "chemical" proton which is incorrect.
RESPONSE: We have coined the term “chemical” proton to specify that, unlike all other protons, it is generated in the chemical reaction of PPi hydrolysis. This definition is now given in line 63. The quotation marks used show that the word chemical is being used in a special way and may not obey the usage rules for the same word without the quotation marks.
- Lines 173-175. The membrane vesicles produced by the procedure used were predominantly inverted (> 90 %), as estimated from the activating effect of alamethicine on their NADH oxidase activity [31] (data not shown). Why is this not shown?
RESPONSE: The requested alamethicine data have been added to Supplement as Table S1; the other supplemental table was renumbered accordingly.
- Kinetics parameters for Table 1 must be previously explained or mentioned in other sections.
RESPONSE: Please note that all parameters in Scheme 1 and Table 1 are explained in the paragraph preceding Table 1. To further clarify Table 1, we have moved Scheme 1 from Materials and Methods to section 2.4 of Results and placed it before Table 1.
- Quality of the figures and tables is very good and easy to follow.
- Authors do not include conclusions.
RESPONSE: We are aware that this section is not obligatory in the IJMS.
Reviewer 3 Report
Comments and Suggestions for Authors
The authors deal with the coupled transport mechanism of Na+ and H+ by the Bacteroides vulgatus Na+,H+-Ppase (Bv-mPPase). Using a clever method of combination of Na+ or H+ ionophores and a pyranine-based H+ concentration dependent fluorescent assay they analyzed the kinetics of the Na+/H+ transport. They found out that Na+ and H+ are transported in an 8:1 relation independent of the Na+ concentration. This interesting observation will help to understand the transport mechanism of the Na+,H+-Ppase. The methods and results of their investigations are clearly and comprehensibly described. However, some explanations and conclusions should be better described to make the results of the study more understandable and comprehensible for those who are not experts in the field of transporter research.
These points are:
1.) “data not shown” is not acceptable. Please show these results at least in the supplement.
2.) L 110: “On the other hand, the prevalence of one activity would favor the transport promiscuity of both subunits, which, for some reasons, exhibits only a marginal dependence on the medium [Na+]/[H+] ratio.”
please, explain in more detail
3.) L 117: “The advantage of pyranine as a pH indicator over the more common acridines is that it reports on changes in pH rather than ΔpH …”
Isn’t change in pH and ΔpH the same?
4.) L 184: “CCCP suppressed the pyranine signal (Figure 3A) as expected, by facilitating H+ exit from the vesicles.”
What is the reason for the remaining fluorescence change?
5.) L 197: “In contrast, the Na+-translocating Da-mPPase caused monensin-sensitive alkalinization of the vesicle lumen in response to PPi addition (Figure 3B) because Na+ accumulation stimulated passive H+ leakage from the vesicles.”
What is the mechanism of the passive H+ leakage?
6.) L 211: “The pyranine response was only partially decelerated by valinomycin, which collapsed Dy.
Is this a quantitative effect, i.e. the massive Na+ influx (which drives the H+ outflux by membrane depolarization) can only partially compensated by the K+ outflow? In other words, what is the driving force for the alkalinization if it is not the membrane potential?
Or does a decrease of the intravesicular K+ concentration limit the hyperpolarizing effect of K+ outflow through the valinomycin transporter?
7.) L 212: “These findings were the first to indicate a marked prevalence of the Na+-translocating activity of Bv-mPPase over its H+-translocating activity—zero signal would be observed if the activities were equal.”
Please explain in more detail.
8.) L230: “The effects of ETH 157 and valinomycin under the conditions shown in Figure 4 were insignificant (< 5%), CCCP stimulated the hydrolysis reaction by only 20%.”
Please, show the data
9.) L273: “Rates of cation transport at 1, 10, and 50 mM Na+ concentrations, as estimated from the data in Figure 5”
Were the rates estimated or rather calculated?
10.) L 279: “The rate measured at 10 mM Na+ concentration (295 nmol· min-1·mg-1) was taken as 100%”
Why? In Fig. 6C are no percentages given.
11.) L 280: “The line shows the best fit of a simple hyperbola to the hydrolysis rate profile.”
Please give the equation.
12.) L287: “… the transport activities required higher Na+ concentrations for saturation (Figure 6A), suggesting a requirement for an additional, lower-affinity Na+-binding site.”
Please explain in more detail.
13.) L 312: “Fitting Equation 4 to the rate dependence”
equation 3
12.) L 351: “These data also rule out the model(s) in which a particular transport function is associated with only one of these complexes.”
Please provide the corresponding reference(s).
13.) L 434: “Accordingly, a constant ratio of the H+ and Na+ transport rates is expected at high Na+ concentrations.”
Give a (estimated) number for high Na+ concentrations.
14.) Paragraph starting with L 453: Please make clearer how model C explains how the Na+/H+ stoichiometry of 8:1 gets independent of the Na+ concentration. A cartoon would be helpful.
15.) L 513: Which device (flame photometer) from which company was used?
16.) L 530: Please explain the rationale of the equations or give the corresponding reference.
17.) L 537: “Cooperative kinetics of PPi hydrolysis was analyzed in terms of Scheme 1 [27] …”
Ref. [27] does not contain scheme 1. Please give the correct reference.
18.) The supplement is missing.
Author Response
We thank the reviewer for the insightful and detailed comments. All are addressed in the revised version or commented as detailed below. Please note that the report contains duplicate items 12 and 13. Modified or added text is highlighted in the revised manuscript.
1.) “data not shown” is not acceptable. Please show these results at least in the supplement.
RESPONSE: The data describing ionophore effects on Bv-mPPase activity and alamethicine effects on NADH oxidase activity have been added to Supplement as Figure S1 and Table S1, respectively; the other supplemental figure and table were renumbered accordingly.
2.) L 110: “On the other hand, the prevalence of one activity would favor the transport promiscuity of both subunits, which, for some reasons, exhibits only a marginal dependence on the medium [Na+]/[H+] ratio.”
please, explain in more detail
RESPONSE: We have simplified this sentence by omitting its second part, keeping in mind that this issue is considered in more detail in the second half of Discussion.
3.) L 117: “The advantage of pyranine as a pH indicator over the more common acridines is that it reports on changes in pH rather than ΔpH …”
Isn’t change in pH and ΔpH the same?
RESPONSE: Not quite so. Pyranine senses pH in the aqueous phase (inside the vesicles in our case), whereas acridines sense ΔpH across the membrane. This is because pyranine is found in the aqueous phase, whereas more hydrophobic acridines are in the lipid membrane. This is now indicated clearer in the revised text (lines 118-119).
4.) L 184: “CCCP suppressed the pyranine signal (Figure 3A) as expected, by facilitating H+ exit from the vesicles.”
What is the reason for the remaining fluorescence change?
RESPONSE: CCCP mediates proton transport across membrane only in the presence of at least a small proton potential difference. The residual proton transport activity observed in the presence of CCCP allows thus keeping on the transport reaction under steady-state conditions.
5.) L 197: “In contrast, the Na+-translocating Da-mPPase caused monensin-sensitive alkalinization of the vesicle lumen in response to PPi addition (Figure 3B) because Na+ accumulation stimulated passive H+ leakage from the vesicles.”
What is the mechanism of the passive H+ leakage?
RESPONSE: It is generally believed that the major pathway for passive proton leakage from the membrane vesicles prepared from mitochondria and bacteria involves the factor Fo of FoF1-ATP synthase, which partially lost F1 during vesicle preparation [Lee C.P., Ernster L. Biochem Biophys Res Commun. 1965, 18:523-529. doi: 10.1016/0006-291x(65)90785-0.]. Because this issue is outside the scope of our study, we have not added this reference.
6.) L 211: “The pyranine response was only partially decelerated by valinomycin, which collapsed Dy.
Is this a quantitative effect, i.e. the massive Na+ influx (which drives the H+ outflux by membrane depolarization) can only partially compensated by the K+ outflow? In other words, what is the driving force for the alkalinization if it is not the membrane potential?
Or does a decrease of the intravesicular K+ concentration limit the hyperpolarizing effect of K+ outflow through the valinomycin transporter?
RESPONSE: We do believe that the transmembrane electric potential difference is the only driving force for the alkalinization. The scenario can be envisaged as follows: the active transport of sodium ions into the vesicles in the presence of valinomycin results in equivalent counterflow of potassium ions, which rapidly decreases their concentration inside the vesicles because of their very low internal volume (approximately 0.2 uL/mg protein). As a consequence, a transmembrane gradient of potassium concentration arises, which forms a diffusional transmembrane potential difference (positive charge inside) due to valinomycin presence. It is this membrane potential difference that drives protons out of the vesicles, causing interior alkalinization. Importantly, this will work only with membrane systems of very low internal volume.
7.) L 212: “These findings were the first to indicate a marked prevalence of the Na+-translocating activity of Bv-mPPase over its H+-translocating activity—zero signal would be observed if the activities were equal.”
Please explain in more detail.
RESPONSE: This is because the two activities change internal pH and, hence, pyranine signal in opposite directions. To address this remark, the sentence has been modified as follows: “… if the activities, which change internal H+ concentration in opposite directions, were equal.”
8.) L230: “The effects of ETH 157 and valinomycin under the conditions shown in Figure 4 were insignificant (< 5%), CCCP stimulated the hydrolysis reaction by only 20%.”
Please, show the data
RESPONSE: The requested data have been added as Figure S1 to the Supplement.
9.) L273: “Rates of cation transport at 1,10, and 50 mM Na+ concentrations, as estimated from the data in Figure 5”
Were the rates estimated or rather calculated?
RESPONSE: Yes, “calculated” is a better word. Replaced.
10.) L 279: “The rate measured at 10 mM Na+ concentration (295 nmol· min-1·mg-1) was taken as 100%”
Why? In Fig. 6C are no percentages given.
RESPONSE: This sentence came from a previous version of Fig 6C by error and has been deleted in the revised version.
11.) L 280: “The line shows the best fit of a simple hyperbola to the hydrolysis rate profile.”
Please give the equation.
RESPONSE: We have further specified in figure legend that the hyperbola was a Michaelis-Menten-type equation.
12.) L287: “… the transport activities required higher Na+ concentrations for saturation (Figure 6A), suggesting a requirement for an additional, lower-affinity Na+-binding site.”
Please explain in more detail.
RESPONSE: If two different methods yield different binding constants for the same ligand, they likely measure binding to different sites. We believe this is clear as is.
13.) L 312: “Fitting Equation 4 to the rate dependence”
equation 3
RESPONSE: The typo has been corrected.
12.) L 351: “These data also rule out the model(s) in which a particular transport function is associated with only one of these complexes.”
Please provide the corresponding reference(s).
RESPONSE: These are hypothetical models. Perhaps, the misunderstanding was caused by the unnecessary “the” before “models”. We have omitted “the” in the revised version and added “hypothetical”.
13.) L 434: “Accordingly, a constant ratio of the H+ and Na+ transport rates is expected at high Na+ concentrations.”
Give a (estimated) number for high Na+ concentrations.
RESPONSE: In principle, this ratio may have any value, and this is now indicated in lines 443–444.
14.) Paragraph starting with L 453: Please make clearer how model C explains how the Na+/H+ stoichiometry of 8:1 gets independent of the Na+ concentration. A cartoon would be helpful.
RESPONSE: The independence of the H+/Na+ transport ratio on Na+ concentration in Model C may have same explanation as in Models A or B. This text has been added in lines 468–470.
15.) L 513: Which device (flame photometer) from which company was used?
RESPONSE: The information on the manufacturer has been added.
16.) L 530: Please explain the rationale of the equations or give the corresponding reference.
RESPONSE: It is now mentioned that these are simple equations for first-order kinetics (line 540).
17.) L 537: “Cooperative kinetics of PPi hydrolysis was analyzed in terms of Scheme 1 [27] …”
Ref. [27] does not contain scheme 1. Please give the correct reference.
RESPONSE: We actually meant reference 30. Corrected.
18.) The supplement is missing.
RESPONSE: When uploading the revised version, we will doublecheck that the supplement is on the journal site.
Round 2
Reviewer 2 Report
Comments and Suggestions for Authors
All the comments and responses are ok.
Author Response
We thank the reviewer for his/her positive evaluation of our manuscript.